# Electricity from methane by reversing methanogenesis

Michael J. McAnulty[1,*], Venkata G. Poosarla[1,*], Kyoung-Yeol Kim[2], Ricardo Jasso-Chávez[3], Bruce E. Logan[2] & Thomas K. Wood[1,4]

Given our vast methane reserves and the difficulty in transporting methane without substantial leaks, the conversion of methane directly into electricity would be beneficial. Microbial fuel cells harness electrical power from a wide variety of substrates through biological means; however, the greenhouse gas methane has not been used with much success previously as a substrate in microbial fuel cells to generate electrical current. Here we construct a synthetic consortium consisting of: (i) an engineered archaeal strain to produce methyl-coenzyme M reductase from unculturable anaerobic methanotrophs for capturing methane and secreting acetate; (ii) micro-organisms from methane-acclimated sludge (including *Paracoccus denitrificans*) to facilitate electron transfer by providing electron shuttles (confirmed by replacing the sludge with humic acids), and (iii) *Geobacter sulfurreducens* to produce electrons from acetate, to create a microbial fuel cell that converts methane directly into significant electrical current. Notably, this methane microbial fuel cell operates at high Coulombic efficiency.

[1] Department of Chemical Engineering, The Pennsylvania State University, University Park, Pennsylvania 16802-4400, USA. [2] Department of Civil and Environmental Engineering, The Pennsylvania State University, University Park, Pennsylvania 16802-4400, USA. [3] Department of Biochemistry, National Institute of Cardiology, Mexico City 14080, Mexico. [4] Department of Biochemistry and Molecular Biology, The Pennsylvania State University, University Park, Pennsylvania 16802-4400, USA. * These authors contributed equally to this work. Correspondence and requests for materials should be addressed to T.K.W. (email: tuw14@psu.edu).

Electrical current may be generated through biological means from organic substrates in microbial fuel cells (MFCs) in which microbes transfer electrons extracellularly (exoelectrogens)[1,2]. In MFCs, exoelectrogens deposit electrons on an anode, from which the electrons migrate to the cathode to create an electrical current; at the cathode, the electrons are passed to another electron acceptor. In a typical dual-chamber set-up, a proton-conducting membrane is used between the anode and cathode chambers to complete the circuit by allowing excess positive charge in the anode (in the form of protons) to migrate to the cathode[3] (Supplementary Fig. 1). Use of MFCs has led to the successful generation of electrical current from a wide variety of substrates, from both natural and artificial sources (including acetate[4], glucose[5], and natural[6] and artificial[7] wastewaters)[8]. However, with methane as a substrate, only negligible electrical power has been reported using uncultured anaerobic methane-oxidizing consortia isolated from oceanic sediment[9,10].

Most studies related to MFCs involve natural exoelectrogens that make electrically conductive pili, such as *Geobacter sulfurreducens*[11,12], or that secrete their own redox-active electron shuttles, such as *Shewanella oneidensis*[13]. Even organisms not naturally capable of external electron transfer may generate electrical current through the use of externally supplied electron shuttling molecules, such as thionine[14], neutral red[14] or methylene blue[15]. However, these exogenously supplied mediators may be toxic and expensive[16].

The potential substrate for MFCs, methane, is also a potent greenhouse gas that can trap heat 38 times more efficiently on a molar basis than carbon dioxide over a span of 20 years[17,18]. Methane extraction from shale deposits has gained considerable attention as methane is an energy dense fuel with decreased carbon dioxide emissions per unit energy[19]. However, procedures of extraction, supply and storage lead to leaks of methane into the atmosphere, with ~8% of the methane produced from a well lost to leaks; up to half of this leaking is due to downstream processes of distribution, transportation, and storage[18]. Conversion of methane to other biofuels, such as through the Fischer-Tropsch process that yields higher molecular weight hydrocarbons, is well established in industry, but large initial capital investments (on the order of $20 billion) are needed for these chemical processing plants[20]. Therefore, processes to convert methane biologically to electricity at the source, if developed, could reduce leaks (from distribution, transportation and storage) and capital cost. Furthermore, harnessing methane has been recognized as one of the most important near-term goals for biochemical engineering[21].

Critically, anaerobic methanotrophs (ANME) participate in anaerobic oxidation of methane (AOM) in nature to play an important role in curbing methane emissions, with roughly 75 Tg of methane each year (88% of methane from natural sources of leaks) in the ocean consumed[22]. AOM in nature typically involves consortia of archaeal ANME paired with sulfate reducing bacteria[23,24] where surplus electrons from methane catabolism are likely transferred between syntrophs directly via multi-haem cytochromes[25] with the terminal electron acceptor (including sulfate[24], nitrate[26], ferric iron[27] and manganic manganese[27]) being replaceable by others such as humic acids and anthraquinone-2,6-disulfonate (AQDS)[28]. However, since no individual species has been isolated that participates in AOM, little is known about it[28,29].

We[30] recently reversed methanogenesis to allow for growth on methane as the main carbon source by engineering the archaeal strain *Methanosarcina acetivorans* to produce methyl-coenzyme M reductase (Mcr) from ANME in an unculturable microbial mat from the Black Sea[31]; the engineered archaeal strain passes electrons to external $Fe^{3+}$ (in the form of $FeCl_3$) and converts methane to acetate[30]. In a follow-up study, we further metabolically engineered the methane-consuming *M. acetivorans* strain to synthesize lactate by adding the gene for the production of a 3-hydroxybutyryl-CoA dehydrogenase from *Clostridium acetobutylicum*[32]. Furthermore, the archaeal host, *M. acetivorans*, was adapted to small pulses of oxygen (named air-adapted *M. acetivorans* (AA)[33]), making it more robust in terms of biofilm production for potential use in a MFC. Here, we combine the metabolically engineered *M. acetivorans* that produces Mcr and grows on methane[30] with *G. sulfurreducens* and methane-consuming sludge to generate substantial electrical current from methane.

## Results

**Acclimation of sludge samples to methane.** Given our goal of creating an MFC that utilizes methane, we desired a consortium that would oxidize methane to provide electrons for the anode via pili[11,12] or multi-haem complexes (possibly associated with other electron shuttles)[34]. We focused on using a consortium since most environmental processes occur within consortia[35], and natural anaerobic methane oxidation involves consortia[23].

To begin formulating our consortium for oxidizing methane, sludge samples from a local anaerobic digester for treating wastewater were acclimated over 567 days to methane as the main carbon source via three successive culturing cycles in high salts (HS) liquid medium with various terminal electron acceptors (Supplementary Table 1). This led to consortia that could be inoculated into methane MFCs for possibly aiding electricity production with electron transfer to the anodes. These cultures were visualized via transmission electron microscopy to reveal cells (both rod-shaped and irregular cocci) attached to each other with pili and subcellular structures (diameter <50 nm) to form biofilm networks (Supplementary Fig. 2); hence, the consortium held promise for conducting electrons in a MFC.

**Electricity from methane.** MFCs were first inoculated with the air-adapted *M. acetivorans* strain[33] that produces ANME Mcr via pES1-MATmcr3 (Supplementary Table 2) so that methane could be converted to acetate[30]. The *M. acetivorans* strain hosting pES1-MATmcr3 can transfer electrons to ferric iron[30]; here, the electrons were consumed by converting ferricyanide to ferrocyanide at the cathode to complete the circuit for the generation of electricity[3]. Although the air-adapted strain can tolerate oxygen and oxidize methane to produce acetate (Supplementary Fig. 3), the whole apparatus was operated anaerobically to eliminate oxygen as a terminal electron acceptor that competes with the generation of an electrical current. *G. sulfurreducens* was added to potentially catabolize acetate to produce electricity[36]. Several days after adding *G. sulfurreducens*, methane-acclimated sludge was added to the MFCs to provide other potential consortial members to produce electricity from methane and other by-products of methane catabolism.

Voltages between the anode and cathode across a fixed $1\,k\Omega$ resistance increased to significant levels (ca., 0.6 V with current density $273\,mA\,m^{-2}$) a few days after sludge addition for MFCs containing both the air-adapted *M. acetivorans* strain producing Mcr and *G. sulfurreducens* (Fig. 1a, Table 1) while the absence of sludge led to no appreciable current (Fig. 1b); these results show that sludge is necessary to generate electricity from methane. *G. sulfurreducens* is also required for electricity, since there is little current in its absence (Fig. 1c, Table 1). Compared to wild-type *M. acetivorans*, the air-adapted strain was also a better host (Fig. 1d, Table 1). Furthermore, there was no current without the air-adapted *M. acetivorans* strain producing Mcr (Fig. 1e) or with only the

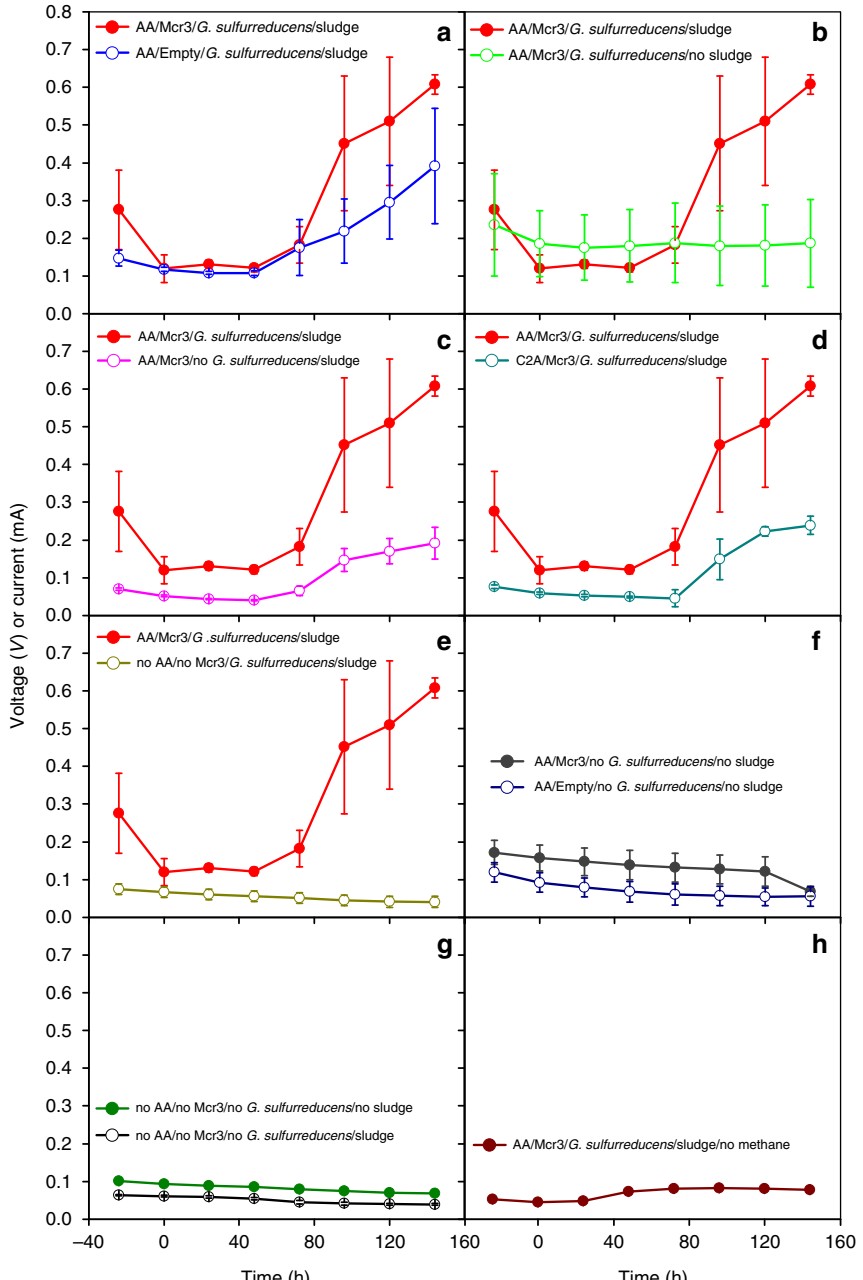

**Figure 1 | Voltages generated in the MFCs with combinations of micro-organisms.** Micro-organisms in the MFCs include combinations of the following: the air-adapted *M. acetivorans* host containing pES1-MAT*mcr3* ('AA/Mcr3'), the air-adapted *M. acetivorans* host containing the empty plasmid pES1(Pmat) ('AA/Empty'), *M. acetivorans* C2A containing pES1-MAT*mcr3* ('C2A/Mcr3'), *G. sulfurreducens* and sludge. The absence of the air-adapted *M. acetivorans* strain is indicated as 'no AA/no Mcr3,' the absence of *G. sulfurreducens* is indicated as 'no *G. sulfurreducens*', the absence of sludge is indicated as 'no sludge' and the replacement of the methane headspace with a nitrogen headspace is indicated as 'no methane'. Sludge was added to the indicated MFCs once the voltage of each MFC decreased to a threshold value of 150 mV or below, at time 0. For MFCs not including sludge, time 0 here is indicated as 132 h after inoculation and set-up. All values (voltages between the anode and cathode across a 1 kΩ fixed resistance) are represented as means ± s.e.m. from at least three replicate MFCs. (**a**–**e**) The red voltage values for 'AA/Mcr3/*G. sulfurreducens*/sludge' are repeated to aid in comparing results to values for (**a**) 'AA/Empty/G. sulfurreducens/sludge', (**b**) 'AA/Mcr3/*G. sulfurreducens*/no sludge', (**c**) 'AA/Mcr3/no *G. sulfurreducens*/sludge', (**d**) 'C2A/Mcr3/*G. sulfurreducens*/sludge' and (**e**) 'no AA/no Mcr3/*G. sulfurreducens*/sludge'. (**f**) Values from 'AA/Mcr3/no *G. sulfurreducens*/no sludge' is compared to values from 'AA/Empty/no *G. sulfurreducens*/no sludge', (**g**) values from 'no AA/no Mcr3/no *G. sulfurreducens*/no sludge' are compared to values from 'no AA/no Mcr3/no *G. sulfurreducens*/sludge', and (**h**) values from 'AA/Mcr3/*G. sulfurreducens*/sludge/no methane' are displayed.

engineered strain (Fig. 1f). Sludge by itself but with methane did not yield electricity (Fig. 1g, Table 1), no current was found without any of the micro-organisms but with methane (Fig. 1g), and without methane, the air-adapted *M. acetivorans* strain producing Mcr, *G. sulfurreducens*, and sludge did not produce current (Fig. 1h).

Therefore, electricity generation from methane depends on the presence of the air-adapted *M. acetivorans* strain producing Mcr, *G. sulfurreducens* and sludge.

Methane consumption followed the voltage generation trends in that MFCs with the air-adapted *M. acetivorans* producing Mcr

**Table 1 | Maximum power generation and current densities in the MFCs.**

| MFC conditions | Maximum power (mW m$^{-2}$) | Maximum current density (mA m$^{-2}$) |
|---|---|---|
| AA/Mcr3/*G. sulfurreducens*/sludge | 168 ± 9 | 273 ± 7 |
| AA/Empty/*G. sulfurreducens*/sludge | 80 ± 70 | 200 ± 100 |
| C2A/Mcr3/*G. sulfurreducens*/sludge | 26 ± 9 | 110 ± 20 |
| AA/Mcr3/no *G. sulfurreducens*/sludge | 20 ± 10 | 90 ± 30 |
| AA/Mcr3/*G. sulfurreducens*/sludge/no methane | 3.0 ± 0.2 | 36 ± 1 |
| no AA/no Mcr3/no *G. sulfurreducens*/no sludge | 4.5 ± 0.4 | 45 ± 2 |
| no AA/no Mcr3/no *G. sulfurreducens*/sludge | 0.2 ± 0.1 | 9 ± 3 |
| AA/Mcr3/no *G. sulfurreducens*/no sludge | 20 ± 20 | 80 ± 40 |
| AA/Empty/no *G. sulfurreducens*/no sludge | 7 ± 5 | 90 ± 50 |

Maximum power and current densities are normalized by the cathode surface area of 0.00227 m$^2$. All MFCs included methane in the headspace unless otherwise indicated. MFCs included at time 0 as indicated the air-adapted *M. acetivorans* host containing pES1-MAT*mcr*3 ('AA/Mcr3'), the air-adapted *M. acetivorans* host containing pES1(Pmat) ('AA/Empty'), *M. acetivorans* C2A containing pES1-MAT*mcr*3 ('C2A/Mcr3'), no *M. acetivorans* strains ('no AA/no Mcr3') and *G. sulfurreducens*. Sludge was added to the MFCs as indicated once the voltage of each MFC decreased to below a threshold value of 150 mV. Averages and s.d.'s between at least three replicates are shown.

paired with *G. sulfurreducens* and sludge having the highest methane consumption (Supplementary Table 3). Methane consumption in MFCs with the air-adapted *M. acetivorans* strain producing Mcr but without both *G. sulfurreducens* and sludge had little methane consumption; hence, electricity is generated as a means to remove the excess electrons from the process for methane oxidation to occur, and the sludge and *G. sulfurreducens* provide a means to conduct the electrons to the anode. Methane losses were not due to leaks, since no oxygen was detected. In MFCs including *G. sulfurreducens* and sludge, production of Mcr from ANME allowed for greater consumption of the methane substrate.

The re-filling of MFCs with more methane (Supplementary Fig. 4) caused an increase in voltage only for MFCs with air-adapted *M. acetivorans* producing Mcr from ANME with *G. sulfurreducens* and methane-acclimated sludge. Corroborating the methane consumption results, production of Mcr from ANME is required for sustained production of electrical current (with a threefold increase in voltage compared to no production of Mcr from ANME, Supplementary Fig. 4). Production of Mcr also increases maximum power output (Table 1). Acetate was confirmed to be used as an intermediate in the methane-to-electricity process, since acetate was not detected in MFCs with air-adapted *M. acetivorans* producing Mcr from ANME with *G. sulfurreducens* and methane-acclimated sludge with a methane headspace after 40 days of incubation (results not shown), while acetate is the main product of methane consumption by air-adapted *M. acetivorans* producing Mcr from ANME (Supplementary Fig. 3).

**Visualization of the MFC anodes**. To determine the presence of potential biofilm structures, MFC anode fibres were visualized by scanning electron microscopy (SEM). Only individual cells could be seen attached to the anode (Supplementary Fig. 5), suggesting that electricity was generated through indirect contact (via electron shuttle molecules) rather than direct contact. The absence of such cell-like structures in the MFCs not inoculated with cells indicates that these structures are cells. The absence of distinctly rod-shaped bacteria attached to the anodes indicated that *G. sulfurreducens* cells, where included, did not attach to the anode. *M. acetivorans* producing Mcr from ANME develops biofilms on FeCl$_3$ precipitates when grown with methane as the substrate[30]; however, the carbon fibre anode surfaces (Supplementary Fig. 5) are much smoother and more structured than the FeCl$_3$ precipitates (Supplementary Fig. 6), so it is likely more difficult for *M. acetivorans* strains to attach to the anodes.

**Sludge replacement by electron shuttles**. Due to the absence of a robust biofilm, we postulated that the bulk of the electron transfer was unlikely due to direct electron transfer but instead occurred via electron shuttles provided by the sludge. To test this hypothesis, we replaced the sludge with five sources of electron shuttles (0.5 mM riboflavin 5′-monophosphate (FMN)[37], 0.5 mM flavin adenine dinucleotide[37], 5 mM AQDS[28], 0.5% humic acids[28] and supernatants from methane-acclimated sludge) and found that of these tested electron shuttles, only humic acids (a complex mix of acids containing carboxyl and phenolate groups) could substitute for the sludge component in MFCs containing the air-adapted *M. acetivorans* strain with pES1-MAT*mcr*3 and *G. sulfurreducens* (Supplementary Fig. 7).

**Characterization of the methane-acclimated sludge**. To determine which micro-organisms were likely active in the methane-acclimated sludge, 16S rDNA analyses were made for the original sludge samples, for the sludge after methane acclimation, and for the sludge used both with MFCs producing and not producing electricity (that is, MFCs with no methane). Critically, the prevalent micro-organism belonged to the genus *Paracoccus* (Fig. 2), and *Paracoccus denitrificans* can significantly substitute for the sludge component in the methane MFCs (Supplementary Fig. 7). As internal controls, both *M. acetivorans* and *G. sulfurreducens* were detected as substantial community members in the MFCs where they were added as part of the synthetic consortium (Fig. 2). The omission of methane (and lack of significant electrical current) did not lead to notable changes in community composition (Fig. 2) which indicates that not many members proliferated substantially under the MFC conditions; however, the relative metabolic activity of each genus member remains to be shown. As expected, the acclimation of the sludge samples to methane decreased the overall diversity (Fig. 2).

**Discussion**

We demonstrate here the generation of significant electrical current using methane as a substrate in MFCs. We also show that our breakthrough technology for extracting electrical power from methane yields a Coulombic efficiency of 90 ± 10% (in MFCs with the air-adapted *M. acetivorans* producing Mcr, *G. sulfurreducens*, and methane-acclimated sludge). Although this Coulombic efficiency is high, it is still reasonable, as other Coulombic efficiencies from MFCs at 90% (ref. 38) and 85% (ref. 39) have been reported. The high Coulombic efficiency of this system, compared to other systems where oxygen from the cathode can be used, or other electron acceptors may be present

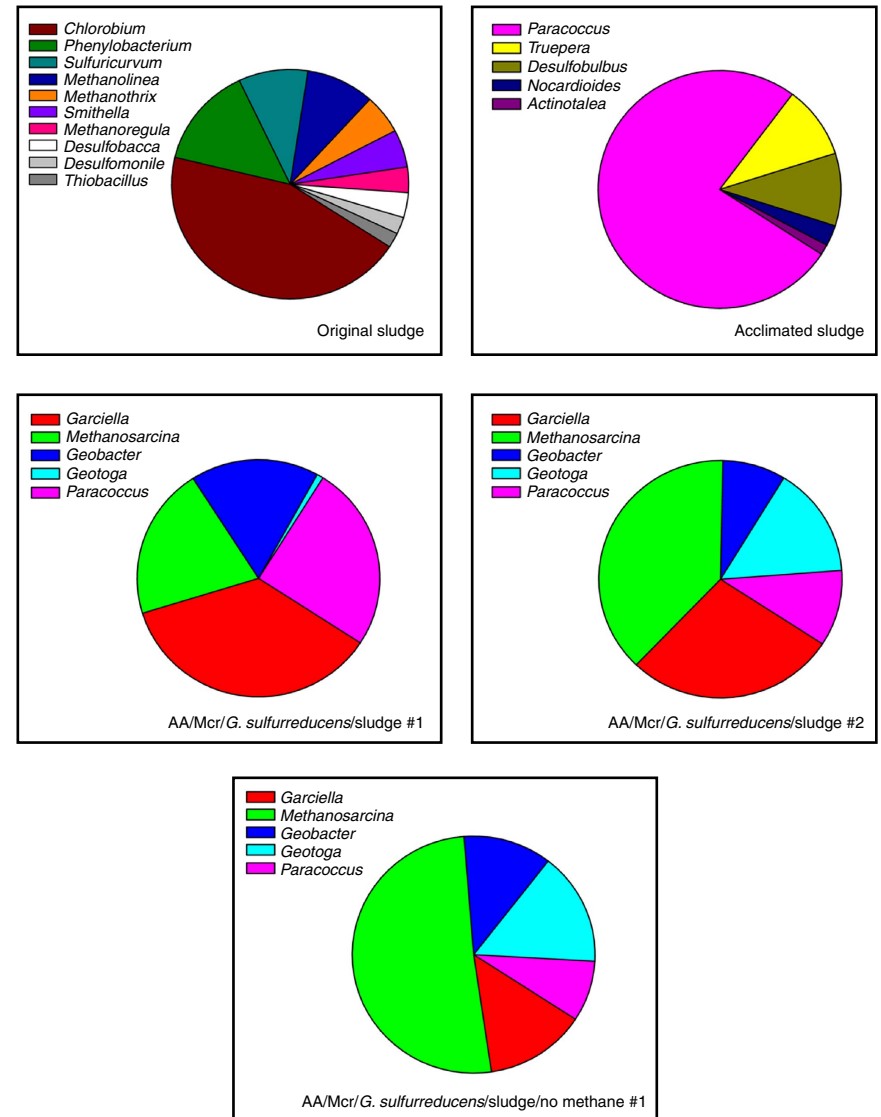

**Figure 2 | Genera identified in sludge and in the MFCs.** Relative compositions are displayed based on the number of identified 16S rDNA reads compared to the total reads and are characterized to the genus taxonomic level. Genera consisting of at least 1% of the relative compositions of each sample are displayed, and DNA reads belonging to these genera are considered for the total reads in each sample. 'Original sludge' is the initial sludge 811 days after isolation, and the 'Acclimated sludge' is the pooled seven Round 3 methane-acclimated sludge cultures (Supplementary Table 1) after 567 days of incubation, with both of these sludge samples used to inoculate MFCs. MFCs included the air-adapted *M. acetivorans* host containing pES1-MAT*mcr*3 ('AA/Mcr3') and *G. sulfurreducens*. Sludge was added to the indicated MFCs once the voltage of each MFC decreased to a threshold value of 150 mV or below. All MFCs included methane in the headspace except the 'AA/Mcr3/*G. sulfurreducens*/sludge/no methane' sample. Samples from MFCs were taken after 40 or 65 days of incubation for the two 'AA/Mcr3/*G. sulfurreducens*/sludge' samples, or after 16 days of incubation for the 'AA/Mcr3/*G. sulfurreducens*/sludge/no methane' sample.

in the solution, reflects the lack of other suitable electron acceptors in the medium other than the anode.

The electricity was generated by a synthetic consortium that includes known species (the air-adapted *M. acetivorans* strain engineered to produce ANME Mcr and *G. sulfurreducens*) combined with either partially characterized species from methane-acclimated sludge isolates or exogenous electron shuttles in the form of humic acids. We also could produce electricity by using only known species, that is, with the sludge replaced by *P. denitrificans*. Hence, the primary role of the sludge was to provide soluble electron shuttles that function like humic acids which are known to transport electrons[28]. The most prevalent genus in the methane-acclimated sludge, *Paracoccus* sp., has demonstrated enhancement of electrical power generation[40],

and many *Paracoccus* spp. are found in soil[41–43]. Although *G. sulfurreducens* is capable of forming electrically conductive biofilm structures[11,12] and the sludge samples acclimated to methane consumption here could form biofilm structures, the bulk of the electron transfer appears to be via electron shuttles. *Geobacter* species are well known to use humic acids and other quinone analogues for electron transfer[44–48], and *G. sulfurreducens* readily transfers electrons to humic acids (and other similar electron acceptors) using multi-haem cyto-chromes and consumes acetate[49,50], so *G. sulfurreducens* plays a role in transferring electrons to the electron shuttles. Hence, our results corroborate the identification of *Geobacter* in MFCs in a previous study that attempted to use methane as a substrate in MFCs[10]. We postulate then that in the MFC with the synthetic

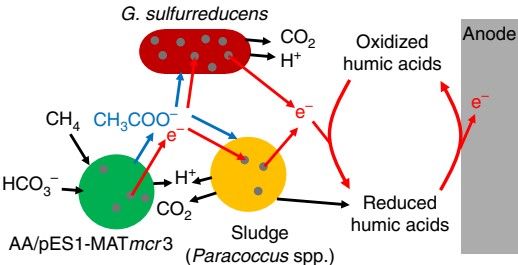

**Figure 3 | Proposed model of biological production of electricity from methane in MFCs.** The air-adapted strain producing Mcr from ANME ('AA/pES1-MAT*mcr*3') consumes methane to produce acetate and excess electrons. Acetate is further oxidized completely to carbon dioxide, producing more excess electrons by *G. sulfurreducens* or sludge that can be substituted to a significant extent by *P. denitrificans* ('Sludge (*Paracoccus* spp.)'). The sludge has a second role in providing humic acid-like electron shuttles, and *G. sulfurreducens* has a second role in providing optimized electron transfer to electron shuttles via multi-haem cytochromes (grey circles) that are located in the outer membrane or S-layer in bacteria and archaea, respectively.

consortium, the ANME Mcr-producing *M. acetivorans* consumes methane to produce oxidized intermediates (including acetate) that are consumed by the *G. sulfurreducens* and sludge components and that the Mcr-producing *M. acetivorans* generates electrons; *G. sulfurreducens* uses multi-haem cytochromes to rapidly transfer electrons to shuttles produced by the sludge or by *P. denitrificans* (Fig. 3).

## Methods

**Bacterial/archaeal strains and cultivation conditions.** The *M. acetivorans* strains (Supplementary Table 1) were routinely grown anaerobically as pre-cultures at 37 °C in an 80% $N_2$/19% $CO_2$/1% $H_2$ atmosphere with mild shaking in 10 ml HS medium[51] or HSYE (HS medium with 2.5 g l$^{-1}$ yeast extract) with 125 mM methanol as the carbon source, unless otherwise indicated. All 28-ml culture tubes (18 × 150 mm, Bellco Glass, Vineland, NJ, USA) were sealed by aluminium crimp seals. pES1-based plasmids were maintained in *M. acetivorans* with 2 µg ml$^{-1}$ puromycin, and methane served to induce ANME Mcr production[30] in the MFCs. Mcr. *G. sulfurreducens* PCA was routinely grown anaerobically in Geobacter Basal medium[52] with 10 mM sodium acetate as the electron donor and 40 mM sodium fumarate as the electron acceptor[53]. Before addition to the MFCs, *P. denitrificans* DSM413 was cultivated aerobically in nutrient broth at 30 °C in a shaker with constant agitation (250 r.p.m.) for 48 h (ref. 54).

For testing oxygen tolerant (ca., 5 days) consumption of methane for the *M. acetivorans* air-adapted strain with high cell-density inocula[30], 2 ml of the strain was pre-grown in 200 ml of HS medium with 125 mM methanol (and 2 µg ml$^{-1}$ puromycin when plasmids were present) at 37 °C for 5 days (turbidity at 600 nm ∼1.0). Cells were collected by centrifugation (5,000 r.p.m. for 20 min), then were washed three times with HS medium and puromycin alone to remove residual methanol. The final cell pellet was resuspended using 5 ml of HS medium supplemented with 10 mM FeCl$_3$ and 2 µg ml$^{-1}$ puromycin to yield a density of $4 \times 10^{10}$ CFU ml$^{-1}$. After filling the headspace of each tube with methane, oxygen was added, where indicated, by replacing a portion of the headspace with air to a final concentration of 1% oxygen. The tubes were incubated at 37 °C with shaking at 250 r.p.m. for 5 days.

Anaerobic sludge from treated wastewater was collected (∼10 ml) just above the sedimentation layer of an anaerobic digester (maintained at 35 °C) from the Office of Physical Plant at the Pennsylvania State University, then kept under a headspace of methane at room temperature. The sample was inoculated (1:100) into HS medium under a methane headspace (HS-methane) with varying concentrations of 0.001–100 mM FeSO$_4$ or FeCl$_3$ for 56 days, then subcultures (1:100 inoculation) were made in HS-methane with 0.001 mM FeSO$_4$ or 1 mM FeCl$_3$, respectively, for 176 days, then the subcultures were grown (1:100 inoculation) in HS-methane with 40 mM ferrihydrite, 1 mM FeCl$_3$, 0.001 mM FeSO$_4$ (with or without 5% $H_2$ in the headspace), or 0.001 mM FeCl$_3$. All incubation steps were conducted at 37 °C.

**Transformation of *M. acetivorans*.** All plasmids were transformed into *M. acetivorans* using a liposome-mediated procedure[55] with slight modifications. Cells (2 ml) grown in HS medium or HSYE with 125 mM methanol to a turbidity at 600 nm of 0.2–0.5 were centrifuged and resuspended into 1 ml transformation

buffer (850 mM sucrose and 80 mM sodium bicarbonate, pH 7.4). Plasmid DNA (4 µg) was mixed with 15–25 µl of DOTAP (n-(1-(2,3-dioleoyloxy)propyl)-n,n,n-trimethylammonium methyl-sulfate, Sigma-Aldrich, St Louis, MO, USA) prepared in transformation buffer in a final reaction volume of 50 µl and incubated at 37 °C for at least 15 min to make DNA:liposome complexes. The 1 ml cell resuspension was mixed with the 50 µl DNA:liposome complex and incubated at 37 °C for 4 h. The cells were then inoculated into 10 ml HS or HSYE medium with 125 mM methanol, then after 48 h of incubation, 1 ml of the culture was added to 10 ml selective HSYE medium with 125 mM methanol and 2 µg ml$^{-1}$ puromycin.

**Electron microscopy.** Cell morphology was examined via transmission electron microscopy (FEI Tecnai G2 Spirit BioTwin, Hillsboro, OR, USA) using uranyl acetate-stained cells[56]. Cellular attachments to anodes were visualized by SEM (Sigma VP-FESEM, Zeiss, Oberkochen, Germany). Samples for SEM were fixed by placing carbon fibres from the anode brush in 2.5% glutaraldehyde in 0.1 M sodium phosphate buffer (pH 7.2), and incubating at room temperature for 1 h. The samples were then washed with increasing concentrations of ethanol: 25, 50, 70, 85, 95 and 100% (three times). Samples were dried via critical point drying before visualizing by SEM.

**Microbial fuel cells.** Two-bottle MFC reactors (total volume of each bottle is 155 ml) were constructed[3] as in Supplementary Fig. 1. All MFC-related power and voltage generation results were performed with three replicates. The two bottles with sideports (2.4 cm inner diameter) were clamped together with a treated Nafion 117 proton exchange membrane (Dupont, Wilmington, DE) and one rubber gasket (3.5 cm outer diameter). The Nafion 117 membranes were cut into 4 × 4 cm squares, then pre-treated by heating for 1 h in solutions of 30% hydrogen peroxide, water, 0.5 M sulfuric acid and water. The top of each MFC chamber consisted of a rubber septum (42 cm diameter, $\frac{1}{4}$-inch thick) and plastic cap with a hole (Corning, Corning, NY, USA). All rubber septa had holes drilled in the centre to allow an electrode to pass. For each anode chamber top, a carbon fibre brush electrode (Mill-Rose, Mentor, OH, USA) from carbon fibres (PANEX 35 50 K, Zoltek, St Louis, MO, USA), twisted into two titanium wires[57] 12.7 cm long and heat-treated at 450 °C for 30 min[58], was passed through the hole in the septum so that 2.8 cm of the top protruded, then the hole was closed with epoxy (Loctite, Dusseldorf, Germany). For each cathode chamber top, a carbon cloth circle (38 mm diameter, Fuel Cell Store, Boulder, CO, USA) was attached to a 10-cm-long titanium wire (1.0 mm diameter, Alfa Aesar, Haverhill, MA, USA), then the wire was passed through the hole in the septum so that 3 cm of the wire protruded, and the hole was closed with epoxy.

To inoculate each MFC anode chamber, cells (*M. acetivorans* and *G. sulfurreducens*) from 200 ml cultures each were collected by centrifugation (5,000 r.p.m. for 20 min) and washed three times with HS medium lacking resazurin (HSNR, redox-active resazurin would short-circuit the MFC system). The final cell pellet was resuspended in 100 ml HSNR with puromycin and placed in the anode chamber. 100 ml of catholyte solution (100 mM ferricyanide in 100 mM sodium phosphate, 5.8 mM ammonium chloride and 1.7 mM potassium chloride pH 7.0) was placed into each cathode chamber, and the caps were closed tightly before removing the MFC from the anaerobic atmosphere. The headspace of each anode chamber was filled with methane. The MFCs were incubated at 30 °C and measurements of the voltage differential across the cathode and anode of each MFC was taken over 1 kΩ resistance at a frequency of 0.05 s$^{-1}$ (with data displayed daily) using a 16-channel differential analogue input module (NI 9205, National Instruments, Austin, TX, USA). *P. denitrificans* DSM413 was added to the MFCs by collecting cells from 200 ml cultures by centrifugation (5,000 r.p.m. for 20 min) and washing three times with HSNR anaerobically. The pellet was left inside the anaerobic chamber to de-aerate completely for 1 h, then resuspended in 5 ml HSNR with puromycin and placed in the anode chamber. To inoculate with sludge, 2 ml of each of the seven Round 3 methane-acclimated sludge cultures (Supplementary Table 1) were combined with 4 ml of the initial sludge isolate. This 18 ml of sludge was centrifuged and resuspended in a total of 8 ml HSNR with puromycin, then 2 ml of the resuspension was injected into each anode chamber.

The addition of sludge supernatants was done similarly by combining 2 ml of each of the seven Round 3 methane-acclimated sludge cultures (Supplementary Table 1) with 4 ml of the initial sludge isolate. This 18 ml of sludge was centrifuged, then 4 ml of filter-sterilized supernatant was added to each MFC. Electron shuttles (humic acids and FMN (Sigma-Aldrich), flavin adenine dinucleotide (Alfa Aesar), AQDS (Carbosynth, Berkshire, UK) and supernatants from methane-acclimated sludge) were included where specified by adding the sterile solutions into HSNR with puromycin to the final concentrations specified.

**Gas chromatography and high-pressure liquid chromatography.** For gas chromatography[30], 50 µl of the gas phase was passed through a 60/80 Carboxen-1000 column (4600 × 2.1 mm, Supelco catalogue no. 12390-U) and a thermal conductivity detector on a 6890 N Agilent gas chromatograph. The injector, column and detector temperatures were maintained at 150, 180 and 280 °C, respectively. Carrier gas flow and reference gas flow (nitrogen used for both) were each maintained at 20 ml per min. Gases were identified based on their retention times, and their concentrations were calculated per comparisons with standards.

High-pressure liquid chromatography[30] was conducted for the quantification of acetic acid. Samples were filtered through a 0.22 μm polyvinylidene fluoride membrane before fractionating 60 μl of a 1:6 dilution (in running buffer, 0.0025 M sulfuric acid in water) with a reversed-phase column [Phenomenex Rezex ROA-Organic Acid H + (8%) (300 × 7.8 mm)], employing a Waters 717 autosampler, a model 515 pump, and a 2996 photodiode array detector (absorbance at 210 nm was used). An isocratic flowrate of 0.4 ml per min 0.0025 M sulfuric acid in water was used. Glacial acetic acid (EMD Millipore, catalogue no. AX0073-6) was used as a standard for comparisons. The peak corresponding to acetic acid was identified based on retention time, and concentrations were calculated per comparisons with standards.

**Characterization of sludge consortia.** Genomic DNA was extracted directly from anode chambers using the UltraClean Microbial DNA Isolation Kit (MoBio, Carlsbad, CA, USA). The v3–v4 hypervariable region of 16S rDNA was amplified using primers PRO341-f and PRO801-r (Supplementary Table 4) universal for prokaryotes (both Archaea and Bacteria domains)[59] with Ilumina (San Diego, CA, USA) adaptors attached for downstream processing. Amplicon libraries were further processed and analysed by the Genome Sciences Facility at the Penn State Hershey College of Medicine using the MiSeq (Illumina) platform according to manufacturer specifications.

**Calculations.** The Coulombic efficiency CE, defined as the fraction of electrons recovered from the substrate, was calculated[38] with modifications using equation (1):

$$CE = \frac{100\,I\,t}{e\,n\,F} \tag{1}$$

where $I$ is average current (current from abiotic reactions with methane from MFCs not inoculated with cells and current not from methane from MFCs inoculated with air-adapted *M. acetivorans* producing Mcr from ANME, *G. sulfurreducens* and sludge, without a methane in the headspace subtracted), $t$ is time, $e$ is the moles of electrons from each mole of methane consumed (8 for methane, from the half reaction equations (2) and (3) below), $n$ is the total mol methane consumed, and $F$ is Faraday's constant.

$$4\,CH_4 + 2\,HCO_3^- \rightarrow 3\,C_2H_3O_2^- + 9\,H^+ + 8\,e^- \tag{2}$$

$$C_2H_3O_2^- + 2\,H_2O \rightarrow 2\,CO_2 + 7\,H^+ + 8\,e^- \tag{3}$$

**Data availability.** The raw 16S rDNA sequencing reads for the methane-acclimated sludge, original sludge, MFC with methane replicate 1, MFC with methane replicate 2 and MFC without methane replicate 1 have been submitted to the NCBI SRA database (http://www.ncbi.nlm.nih.gov) under biosample accessions SAMN06562530, SAMN06562531, SAMN06562532, SAMN06562533 and SAMN06562534, respectively. The other data that support the main findings in this study are available from the authors upon reasonable request.

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

## Acknowledgements

This work was supported by the Department of Energy Advanced Research Projects Agency—Energy. We thank Dr. Yuka Imamura Kawasawa and Anna C. Salzburg at The Pennsylvania State University College of Medicine for work with 16S rDNA sequencing and its analysis.

## Author contributions

T.K.W. conceived the project. M.J.M. and V.G.P. designed experiments and interpreted the data. R.J.-C. provided the air-adapted strain of *M. acetivorans*. K.-Y.K and B.E.L. provided insights into the MFC set-up. M.J.M, V.G.P. and T.K.W. wrote the manuscript.

## Additional information

**Competing interests:** The authors declare no competing financial interests.

**Publisher's note**: 

