## [Peer Review File · Nature Communications]

Reviewers' comments:

Reviewer #1 (Remarks to the Author):

Review of "Electricity from Methane by Reversing Methanogenesis"

Authors: McAnulty, M.J. et al.

This manuscript describes experiments with small-scale microbial fuel cells (MFCs) that succeeded in generating electricity coincident with the consumption of methane in the presence of an air-adapted metabolically-engineered strain of *M. acetivorans*, *G. sulfurreducens*, a microbial consortia enriched from wastewater sludge acclimated to methane, and electron shuttle molecules contained either in a waste-water sludge supernatant or a mixture of humic acids. The paper's title refers to the role of the archaeal strain *M. acetivorans* in presumably converting methane to acetate within the anode chamber of the microbial fuel cell. The novelty of the work is the claim that these studies may lead to a commercial process to directly produce electricity from methane and reduce methane losses to the atmosphere during the distribution, transport and storage of natural gas. Research in this area of biochemical engineering is of broad interest.

Overall the paper is clearly written and the conclusions well-supported by the experimental data that includes genomic characterizations and electron microscopy. However, I do not think it is ready yet for publication. Figure 1 (the most critical figure) is difficult to make sense of unless a reader first studies the methods. The caption (or text around line 78) needs to make clear that the voltages displayed are between cathode and anode connected across a fixed resistance of 1000 ohms. Since the authors refer to current production (e.g., at line 83), it would be informative if Figure 1 also displayed current. What would be most insightful would be additional time course measurements linking current generation to changes in methane, acetate, CO₂ (and perhaps H₂) in the anode compartment (solution and/or head space) for the best performing treatments. Supplementary Figure 3 only gives an indication of some of these changes and does not provide a timeframe. It is not clear if the principal anode reactions change over the course of individual MFC experiments or what columbic efficiencies are achieved in the oxidation of methane. Can the authors provide estimates? With such further considerations the study seems sufficiently promising for publication.

Other minor comments are:

Line 21. Can the authors clarify what is meant here by "in an anaerobic process"?

Line 35. Suggest rewrite as "as by the Fischer-Tropsch process that yields higher molecular weight hydrocarbons..."

Line 37. Clarify what is meant by "at the site". Do you mean at the source?

Line 45. The authors are encouraged to include more recent references that show more is being learned about different mechanisms of AOM.

Line 217. Can the authors give more information about the HSNR medium? How important is this to the outcome? Were other media tried?

Line 225. Can the authors explain further how they developed the sludge inoculation procedure and why it was done as reported? It seems complicated. Were alternate procedures tried?

Reviewer #2 (Remarks to the Author):

Methane oxidizing microorganisms have received much attention recently due to their potentials on using the abundant and low cost natural gas. It is interesting the authors converted methane to electricity using microbial fuel cells, but there are also questions:

1. There are many questions regarding the justification of producing electricity from methane for better transportation and storage. It is a consensus that electricity is actually more expensive to transport and store and therefore there are many efforts to convert renewable electricity to gaseous and liquid fuels, not the opposite route this study proposes. Plus, using microbial fuel cell to convert methane to electricity seems too slow, too expensive, and too complicated – why not just burn methane to produce electricity?

2. There have been recent studies used similar approaches to convert methane to electricity in microbial fuel cells, and the reference below used a simple system. Some convincing justifications of the novelty are needed.

Ding, Jing, et al. "Decoupling of DAMO archaea from DAMO bacteria in a methane-driven microbial fuel cell." *Water Research* (2017). 110, 112–119

3. Line 74 – "Although the air-adapted strain can tolerate oxygen and oxidize methane to produce acetate (Supplementary Fig. 3), the whole apparatus was operated anaerobically." - Can you justify why here, how does anaerobic condition affect the strain's metabolism?

4. Figure 1 – The system is quite confusing and complicated. It is not clear what are the mechanisms associated with the pure cultures and the unknown sludge inoculum, and how do they relate to each other and benefited from each other since the authors concluded no methane to electricity conversion can be completed without any of the three players (*M. acetivorans*, *G. sulfurreducens*, and sludge). In theory the two pure cultures can make the conversion but apparently that couple didn't work until a mixed culture sludge or electron shuttles were added. However, a similar thing happens if no *G. sulfurreducens* was present in the system. It is just too confusing and needs much better explanations.

5. Did you measure acetate concentration change to understand its production and consumption? This will help understand the syntrophy.

6. Page 120 – this gets more confusing – *G. sulfurreducens* is known for direct electron transfer without using electron shuttles. Why such molecules were needed in this system? I don't think this has anything to do with *M. acetivorans* as its role is to produce acetate.

Response to the reviewers' comments for manuscript NCOMMS-16-28590-T, "Electricity from Methane by Reversing Methanogenesis"

Summary

We have conducted two new experiments including experiments that (i) determined that the sludge component in our MFCs may be replaced with *Paracoccus denitrificans* (*Paracoccus* spp. was identified by us in the first draft through genomic sequencing to be the dominant member of the sludge microorganisms) which indicates the role of sludge was to provide electron shuttles as this has been demonstrated previously for this strain and (ii) determined that acetate was completely consumed at the end of 40 days of incubation in the MFCs of interest (including the air-adapted strain with *G. sulfurreducens* and sludge). The results of replacing sludge with *P. denitrificans* corroborate our genomic studies that identified *Paracoccus* spp. as important for converting methane into electricity. Furthermore, we have added (i) a schematic which summarizes our conclusions for how current is created and transported in the MFCs (**Fig. 3**) and (ii) an estimate of the Coulombic efficiency that shows our process is extraordinarily efficient ($90 \pm 10\%$). We now also document that *G. sulfurreducens* can transfer electrons not just by direct transfer but by passage to electron shuttles, which cements its importance in our MFC and sheds light on its role in the process. We note that this is the first process that converts methane to significant electricity and so it is quite novel.

We would like to thank the reviewers for their comments and have addressed all of the queries to make the manuscript stronger. The reviewers' remarks follow and our response is underlined. Line numbers refer to the revised version of the manuscript.

Reviewer 1

This manuscript describes experiments with small-scale microbial fuel cells (MFCs) that succeeded in generating electricity coincident with the consumption of methane in the presence of an air-adapted metabolically-engineered strain of *M. acetivorans*, *G. sulfurreducens*, a microbial consortia enriched from wastewater sludge acclimated to methane, and electron shuttle molecules contained either in a waste-water sludge supernatant or a mixture of humic acids. The paper's title refers to the role of the archaeal strain *M. acetivorans* in presumably converting methane to acetate within the anode chamber of the microbial fuel cell. The novelty of the work is the claim that these studies may lead to a commercial process to directly produce electricity from methane and reduce methane losses to the atmosphere during the distribution, transport and storage of natural gas. Research in this area of biochemical engineering is of broad interest.

Overall the paper is clearly written and the conclusions well-supported by the experimental data that includes genomic characterizations and electron microscopy. However, I do not think it is ready yet for publication.

Figure 1 (the most critical figure) is difficult to make sense of unless a reader first studies the methods.

As suggested, details have been added to the caption of Fig. 1 so that the reader can understand it more easily without having to read the Methods first. In addition, we have re-drawn each panel to simplify the display (i.e., replaced the labels for the curves with color-coded legends and show both voltage and current in a single y-axis label).

The caption (or text around line 78) needs to make clear that the voltages displayed are between cathode and anode connected across a fixed resistance of 1000 ohms.

As suggested, we now explicitly state that "Voltages between the anode and cathode across a fixed 1 k Ω resistance increased..." (line 84).

Since the authors refer to current production (e.g., at line 83), it would be informative if Figure 1 also displayed current.

As suggested, Fig. 1 has been modified to also display current.

What would be most insightful would be additional time course measurements linking current generation to changes in methane, acetate, CO₂ (and perhaps H₂) in the anode compartment (solution and/or head space) for the best performing

treatments.

We previously showed formation of acetate from methane (and from bicarbonate) by *M. acetivorans* producing Mcr from ANME¹ and previously showed in the first draft that the air-adapted *M. acetivorans* producing ANME also produces acetate as the main by-product of anaerobic methane oxidation (**Supplementary Fig. 3**). As suggested, we now show that the acetate was completely consumed after 40 days of incubation in our best-performing MFCs (including the air-adapted strain with *G. sulfurreducens* and sludge) (representative analysis chromatogram shown **Fig. R1**, and these results have been added to line 112). Hence, we can conclude that acetate is an intermediate in the overall process of methane consumption for electricity generation. Note we have avoided time-course measurements of the liquid and gas components in the anode chambers to leave the MFCs as undisturbed as possible.

Supplementary Figure 3 only gives an indication of some of these changes and does not provide a timeframe. It is not clear if the principal anode reactions change over the course of individual MFC experiments or what coulombic efficiencies are achieved in the oxidation of methane. Can the authors provide estimates? With such further considerations the study seems sufficiently promising for publication.

We agree with the reviewer that **Supplementary Fig. 3** only gives an indication of some of these changes. However, the main point of **Fig. 3** is to show that the air-adapted *M. acetivorans* strain can produce functional Mcr from ANME (we previously showed this characteristic only with the *M. acetivorans* C2A strain¹) by consuming methane and producing acetate. Changes of principal anode reactions (consisting of transfer of electrons to generate an electrical current) over time are shown in **Fig. 1** and **Supplementary Fig. 4**.

As suggested, we now also report a conservative estimate of the Coulombic efficiency in our best-performing MFC system (inoculated with air-adapted *M. acetivorans* containing pES1-MATmcr3, *G. sulfurreducens*, and methane-acclimated sludge, under a methane headspace) of $90 \pm 10\%$ over a span of 35 days (line 149).

This value was estimated using the following equation:

$$CE = \frac{100 I t}{e n F}$$

where *I* is average current (current from abiotic reactions with methane from MFCs not inoculated with cells and current not from methane from MFCs inoculated with air-adapted *M. acetivorans* producing Mcr from ANME, *G. sulfurreducens*, and sludge, without a methane in the headspace subtracted), *t* is time, *e* is the moles of electrons from each mole of methane consumed (8 for methane, from the half reaction equations below), *n* is the total mol methane consumed, and *F* is Faraday's constant.

The Coulombic efficiency is then approximated as:

Fig. R1. Representative HPLC chromatograms for acetate quantitation/detection. Chromatograms from a sample from a MFC containing the air-adapted strain producing Mcr from ANME, *G. sulfurreducens*, and sludge after 40 days (Original), and spiked with 12 mM acetate (Spiked). Acetate has a retention time of 24 minutes under these conditions.

$$CE = \frac{((0.38 \pm 0.04) - (0.046 \pm 0.004) - (0.063 \pm 0.002))mA * (35 d) * (86400 s/d)}{8 \frac{mol e^-}{mol CH_4} * (1.197 \pm 0.002)mmol CH_4 * 96485 \frac{s * mA}{mmol}} * 100\%$$

Other minor comments are:

Line 21. Can the authors clarify what is meant here by “in an anaerobic process”?

As suggested, we have deleted this unnecessary qualifier as there are no MFCs, aerobic or anaerobic, that have utilized methane to make significant electricity (line 22).

Line 35. Suggest rewrite as “as by the Fischer-Tropsch process that yields higher molecular weight hydrocarbons....”

As suggested, the text has been rewritten for clarification (line 36).

Line 37. Clarify what is meant by “at the site”. Do you mean at the source?

As suggested, we now indicate “at the source” (line 39).

Line 45. The authors are encouraged to include more recent references that show more is being learned about different mechanisms of AOM.

As suggested, we now include more recent references regarding AOM and expand the Introduction accordingly (line 45). We also added additional information^{2,3} to clarify that the role of *G. sulfurreducens* in our system is to provide enhanced electron transfer to humic acids or electron shuttles similar to humic acids (line 163).

Line 217. Can the authors give more information about the HSNR medium? How important is this to the outcome? Were other media tried?

In our experience, the HS medium base provides optimal growth conditions for *M. acetivorans* strains. So, we also used an HS liquid component for acclimating sludge samples for growth on methane. One of the components in HS medium, resazurin, is a redox-active chemical that could short-circuit the MFC system, so HS medium without resazurin (HSNR) was used in the MFCs; this is now clarified (line 237). No other media were tried, so this suggestion is a good idea for our future to optimize our MFC conditions.

Line 225. Can the authors explain further how they developed the sludge inoculation procedure and why it was done as reported? It seems complicated. Were alternate procedures tried?

Initially, sludge was not used, so the microbial fuel cells (MFCs) were first set up with the *M. acetivorans* or *G. sulfurreducens*. Sludge was then added once only insignificant voltages were seen in these initial systems. We decided to keep the sludge inoculation conditions consistent once significant voltages were seen when all three components (air-adapted *M. acetivorans* producing Mcr from ANME, *G. sulfurreducens*, and sludge) were included. This inoculation procedure also allowed for easier visualization of the long-term (lasting at least 6 days) effect of sludge (or any other substituting component including electron shuttles or *Paracoccus denitrificans*) addition on MFC voltages. No alternate procedures for sludge inoculation were tried.

Reviewer 2

Methane oxidizing microorganisms have received much attention recently due to their potentials on using the abundant and low cost natural gas. It is interesting the authors converted methane to electricity using microbial fuel cells, but there are also questions:

1. There are many questions regarding the justification of producing electricity from methane for better transportation and storage. It is a consensus that electricity is actually more expensive to transport and store and therefore there are many efforts to convert renewable electricity to gaseous and liquid fuels, not the opposite route this study proposes. Plus, using

microbial fuel cell to convert methane to electricity seems too slow, too expensive, and too complicated – why not just burn methane to produce electricity?

We agree with the reviewer that transportation of methane to a site may be more efficient than electricity depending on the distances, but the use of methane for electricity production in a non-combustion based process is of interest. For example, taking digester gas at a wastewater treatment plant and converting it to energy in an MFC might be more practical than combustion processes which require a large scale to be practical⁴. In addition, MFC conversion to electricity does not generate combustion gases and therefore the potential formation of harmful nitrogen oxides and sulfur oxides⁵. Our study also grants additional insight into a currently intractable, natural process by giving further credence to the likelihood that some methanotrophic archaea shuttle electrons using multi-heme cytochromes⁶ (*M. acetivorans* produces multi-heme cytochromes⁷) to syntrophs⁸. Our study is also the first known instance of creating a robust MFC that utilizes methane. Other merits include constructing a synthetic consortium (consisting of characterized species, or partly uncharacterized species) for the purpose of anaerobic methane consumption. We now also quantify the importance of our process by demonstrating efficient methane conversion from electricity based on Coulombic efficiency: our MFC operates at $90 \pm 10\%$ in our best-performing MFCs (air-adapted strain with *G. sulfurreducens* and sludge with methane), (line 149). For these reasons, we believe this topic is both of great interest and importance.

2. There have been recent studies used similar approaches to convert methane to electricity in microbial fuel cells, and the reference below used a simple system. Some convincing justifications of the novelty are needed. Ding, Jing, et al. "Decoupling of DAMO archaea from DAMO bacteria in a methane-driven microbial fuel cell." *Water Research* (2017). 110, 112–119.

This manuscript was published while ours was under review⁹. Furthermore, this group⁹ reported insignificant current generation so theirs is not a competing or significant process (they report their maximum voltage at 25 mV, which is 25-fold lower compared to our 620 mV). The power generated in their system was also very low compared to ours (0.653 mW/m², so it was 250-fold lower compared to our 168 mW/m²). Critically, a power density <1 mW/m² can be due to many side reactions and is not sufficient evidence of methane to electricity. The authors of this paper state, "The DAMO fuel cell worked successfully but demonstrated weak electrogenic capability with around 25 mV production."⁹, hence, the work cited by the reviewer not only underscores the novelty of our work but demonstrates how powerful our breakthrough is in terms of obtaining large amounts of methane conversion. The authors also saw an enrichment of *Geobacter*⁹, which confirms our choice of *Geobacter* species for our methane-powered microbial fuel cells.

3. Line 74 – “Although the air-adapted strain can tolerate oxygen and oxidize methane to produce acetate (Supplementary Fig. 3), the whole apparatus was operated anaerobically.” - Can you justify why here, how does anaerobic condition affect the strain’s metabolism?

Aerobic operation presents the possibility for the microbes to use oxygen as a terminal electron acceptor to compete with the generation of electrical current, so it was necessary to use anaerobic conditions. Of the components used in our MFC systems, *G. sulfurreducens* has the capability to use oxygen as a terminal electron acceptor¹⁰, and facultative aerobes possibly existed in the sludge population that could as well. This is now clarified in the manuscript (line 80). We indicated that the air-adapted strain producing Mcr from ANME produces acetate as a product when consuming methane with or without the presence of oxygen (**Supplementary Fig. 3**), so oxygen likely does not induce major metabolic changes on the air-adapted strain during methane consumption.

4. Figure 1 – The system is quite confusing and complicated. It is not clear what are the mechanisms associated with the pure cultures and the unknown sludge inoculum, and how do they relate to each other and benefited from each other since the authors concluded no methane to electricity conversion can be completed without any of the three players (*M. acetivorans*, *G. sulfurreducens*, and sludge). In theory the two pure cultures can make the conversion but apparently that couple didn’t work until a mixed culture sludge or electron shuttles were added. However, a similar thing happens if no *G. sulfurreducens* was present in the system. It is just too confusing and needs much better explanations.

As suggested, we added the schematic of **Fig. 3** to better explain the purpose of each anode component, and we expanded the **Fig. 1** caption to help avoid confusion. *G. sulfurreducens* readily transfers electrons to humic acids^{2, 3}, so it likely provides enhanced electron transfer to humic acids/electron shuttles (produced by the sludge or provided exogenously by us) compared to the *M. acetivorans* and/or sludge (or *Paracoccus*) components, while consuming acetate (produced from

methane by the *M. acetivorans* component) to generate more electrons for transfer (line 161). Mere contact of an oxidized quinone (such as found in humic acids) with a reduced heme group on a c-type cytochrome on the surface of a *G. sulfurreducens* is likely enough to catalyze electron transfer². Also, reduction of insoluble electron acceptors likely requires rearrangements of cytochromes for optimized transfer², explaining why indirect electron transfer to the carbon fiber anodes was seen in our methane MFCs. *G. sulfurreducens* was also enriched in methane-based MFCs⁹, confirming the importance of *G. sulfurreducens* seen here.

5. Did you measure acetate concentration change to understand its production and consumption? This will help understand the syntrophy.

We did not measure acetate concentration changes. But, we had previously shown that acetate was the main by-product of reverse methanogenesis by *M. acetivorans* producing Mcr from ANME¹, and we show here that acetate was also produced by the air-adapted *M. acetivorans* strain producing Mcr from ANME when placed under a methane headspace (**Supplementary Fig. 3**). When the *M. acetivorans* component was not included in MFCs, voltages were negligible (see **Fig. 1E**). We also saw that no acetate was produced in MFCs inoculated with air-adapted *M. acetivorans* producing Mcr from ANME, *G. sulfurreducens*, and sludge under a methane headspace (**Fig. R1**, line 112), and therefore conclude that acetate is the main intermediate passed between the *M. acetivorans* strain and the *G. sulfurreducens* and/or sludge components and that acetate was fully consumed. The use of acetate as an intermediate is now explained in **Fig. 3**.

6. Line 120 – this gets more confusing – *G. sulfurreducens* is known for direct electron transfer without using electron shuttles. Why such molecules were needed in this system? I don't think this has anything to do with *M. acetivorans* as its role is to produce acetate.

We now explain our electron transfer model visually in **Fig. 3**. Although *G. sulfurreducens* is capable of transferring electrons directly to solid surfaces, we did not see significant biofilm-related structures on the anodes when viewed by scanning electron microscopy. We therefore concluded that the majority of electron transfer is through indirect mechanisms, where the sludge component is responsible for providing electron shuttles needed for indirect electron transfer. Critically, beyond direct electron transfer by *G. sulfurreducens*, electron shuttles such as humic acids enhance electron transfer from *G. sulfurreducens* to insoluble electron acceptors². Therefore, the role of the *M. acetivorans* component is to provide acetate and electrons, but further catabolism of acetate by *G. sulfurreducens* in the anode chamber yields more electrons, all of which were deposited on the anode to generate the observed electrical currents. The electron shuttles in the MFC serve to transfer electrons from *G. sulfurreducens* and were thus needed to carry the electrons generated by the catabolism of methane and acetate from cells to the anode.

References

1. Soo, V.W., McAnulty, M.J., Tripathi, A., Zhu, F., Zhang, L., Hatzakis, E., Smith, P.B., Agrawal, S., Nazem-Bokae, H., Gopalakrishnan, S., Salis, H.M., Ferry, J.G., Maranas, C.D., Patterson, A.D. & Wood, T.K. Reversing methanogenesis to capture methane for liquid biofuel precursors. *Microb. Cell Fact.* **15**, 11 (2016).
2. Voordeckers, J.W., Kim, B.C., Izallalen, M. & Lovley, D.R. Role of *Geobacter sulfurreducens* outer surface c-type cytochromes in reduction of soil humic acid and anthraquinone-2,6-disulfonate. *Appl. Environ. Microbiol.* **76**, 2371-2375 (2010).
3. Richter, K., Schicklberger, M. & Gescher, J. Dissimilatory reduction of extracellular electron acceptors in anaerobic respiration. *Appl. Environ. Microbiol.* **78**, 913-921 (2012).
4. Haynes, C.A. & Gonzalez, R. Rethinking biological activation of methane and conversion to liquid fuels. *Nat. Chem. Biol.* **10**, 331-339 (2014).
5. Peighambaroust, S.J., Rowshanzamir, S. & Amjadi, M. Review of the proton exchange membranes for fuel cell applications. *Int. J. Hydrogen Energ.* **35**, 9349-9384 (2010).
6. McGlynn, S.E., Chadwick, G.L., Kempes, C.P. & Orphan, V.J. Single cell activity reveals direct electron transfer in methanotrophic consortia. *Nature* **526**, 531-535 (2015).
7. Li, Q., Li, L., Rejtar, T., Lessner, D.J., Karger, B.L. & Ferry, J.G. Electron transport in the pathway of acetate conversion to methane in the marine archaeon *Methanosarcina acetivorans*. *J. Bacteriol.* **188**, 702-710 (2006).
8. Scheller, S., Yu, H., Chadwick, G.L., McGlynn, S.E. & Orphan, V.J. Artificial electron acceptors decouple archaeal methane oxidation from sulfate reduction. *Science* **351**, 703-707 (2016).

9. Ding, J., Lu, Y.Z., Fu, L., Ding, Z.W., Mu, Y., Cheng, S.H. & Zeng, R.J. Decoupling of DAMO archaea from DAMO bacteria in a methane-driven microbial fuel cell. *Water Res.* **110**, 112-119 (2016).
10. Lin, W.C., Coppi, M.V. & Lovley, D.R. *Geobacter sulfurreducens* can grow with oxygen as a terminal electron acceptor. *Appl. Environ. Microbiol.* **70**, 2525-2528 (2004).

Reviewers' comments:

Reviewer #2 (Remarks to the Author):

Based on previous reviews, the authors made significant improvements on this revised version. There are several concerns that still need clarifications:

1. Abstract and discussion claims – I don't think the authors should claim this is the first study that converts methane directly into current. As previous reviewer commented other groups have reported similar processes though their power density was lower than this one. The authors should give credits to these studies and provide discussions on prior art.
2. Line 100 - Since mixed sludge sample was added in the system, how do you prove no other electron acceptor is present?
3. There are some hypotheses that require further literature or experimental supports. For example, why and how *Geobacter* transferred electrons to the electron shuttles rather than the anode, and how *Geobacter* attempted to use methane as a substrate.
4. Figure 3 – this relay-type of electron transfer really needs further explanation and proof, as it seems contradict to traditional model of extracellular electron transfer by *Geobacter*. If this proposed pathway is true, how *Geobacter* competes with sludge? Which step is rate limiting? Is it possible to locate *Geobacter* in solution or biofilm?
5. Can you comment on methane mass transfer in liquid and solution acidification in the system?
6. While it is hard to repeat the calculation in Coulombic efficiency as no raw data was given, but $90 \pm 10\%$ of CE is very impressive and to be frankly a little bit too good to be true, especially since mixed sludge was added. Will be good to double check every step of this calculation to make sure it is accurate.

Response to the reviewers' comments for manuscript NCOMMS-16-28590A, "Electricity from Methane by Reversing Methanogenesis"

Summary

We have revised the manuscript to as indicated in the point-by-point response below and added color to Fig. 3 to emphasize the electron and acetate paths.

We would like to thank the reviewer for their comments and have addressed all of the queries to make the manuscript stronger. The reviewers' remarks follow and our response is underlined. Line numbers refer to the revised version of the manuscript.

Reviewer 2

1. Abstract and discussion claims – I don't think the authors should claim this is the first study that converts methane directly into current. As previous reviewer commented other groups have reported similar processes though their power density was lower than this one. The authors should give credits to these studies and provide discussions on prior art.

We have already cited all the relevant (and highly limited) literature (refs 9 and 10, line 23) in our previous draft, and we note again that methane was not used successfully to make electricity in these studies since a power density of less than 1 mW/m² can be due to many side reactions and thus a very low current or power density is not sufficient evidence of converting methane to electricity. However, to soften our claim, we have made four changes to the text: (i) line 5 "previously" is replaced with "with much success previously", (ii) line 11 "significant" was added, (iii) line 60 "for the first time" is deleted, and (iv) line 146 "first" is replaced by "significant".

2. Line 100 - Since mixed sludge sample was added in the system, how do you prove no other electron acceptor is present?

As suggested, we have deleted "Since no extra electron acceptor is included in the anode compartment for methane consumption," as there is no need for this statement; clearly electrons must be removed from the cells for them to grow on methane and these electrons are given to the anode. Our results show clearly that in the absence of *G. sulfurreducens* and sludge, the electrons are not making it to the anode and methane is not oxidized. This is now clarified in the manuscript (line 101).

We have also run a number of experiments that support our claim that there is no other electron acceptor or electron donor used, by conducting control experiments. Please see the text (line 92):

"Sludge by itself but with methane did not yield electricity (Fig. 1G, Table 1), no current was found without any of the microorganisms but with methane ((Fig. 1G), and without methane, the air-adapted *M. acetivorans* strain producing Mcr, *G. sulfurreducens*, and sludge did not produce current (Fig. 1H). Therefore, electricity generation from methane depends on the presence of the air-adapted *M. acetivorans* strain producing Mcr, *G. sulfurreducens*, and sludge."

In addition, we ran tests with the sludge successfully replaced by humic acids. We clarified the manuscript to indicate that only the addition of humic acids, and not sludge supernatants (which would contain soluble electron acceptors from the sludge inoculum) could replace the sludge component (line 130):

"...we replaced the sludge with five sources of electron shuttles...and found that of these tested electron shuttles, only humic acids (a complex mix of acids containing carboxyl and phenolate groups) could substitute for the sludge component in MFCs containing the air-adapted *M. acetivorans* strain with pES1-MATmcr3 and *G. sulfurreducens* (Supplementary Fig. 7)."

3. There are some hypotheses that require further literature or experimental supports. For example, why and how *Geobacter* transferred electrons to the electron shuttles rather than the anode, and how *Geobacter* attempted to use methane as a substrate.

Geobacter species are well known to use humic acids and other quinone analogs for electron transfer,¹⁻⁵ so we do not believe we need to further investigate the interactions of *Geobacter* with shuttles. More importantly, recent studies are

now showing electron transfer to methanogens, and so there is some knowledge on interspecies electron transfer between *Geobacter* and methanogens⁶⁻⁸. However, the mechanisms by which interspecies electron transfer occurs is poorly understood, and is just now becoming clearer in the case of *Geobacter* interactions with *Methanosaeta*,⁷ for example. As suggested, to document more fully that *Geobacter* transfers electrons to electron shuttles, these references and explanations have been added (line 164).

Note that *Geobacter* is not claimed to use methane; instead, methane is converted to acetate by the engineered archaeal strain and *Geobacter* uses that acetate (**Fig. 3**).

We have presented this new breakthrough finding of methane used for significant current generation, and we believe it is beyond the scope of this study to *more* fully describe the involved mechanisms (e.g., we have determined the mechanism by which the sludge participates in the engineered consortium, through electron shuttles and have identified the dominant strain in the sludge, *Paracoccus*). Thus, we are not able to further address this point (beyond clarifying the manuscript with additional literature citations), and plan on performing further mechanistic studies in the future.

4. Figure 3 – this relay-type of electron transfer really needs further explanation and proof, as it seems contradict traditional model of extracellular electron transfer by *Geobacter*.

Please see the response to item 3 above documenting it is well-known that *Geobacter* spp. transfer electrons to humic acids and other quinone analogs. We have also shown that (i) there is not significant biofilm formation via scanning electron microscopy (**Supplementary Fig. 5**), (ii) that humic acids may replace sludge as an electron shuttle (**Supplementary Fig. 7**), and (iii) that a strain that facilitates electron transport (*Paracoccus denitrificans*) can replace sludge; therefore, these three lines of evidence show the electron transfer is likely due to electron shuttles. Critically, refs 49 and 50 in the manuscript demonstrate *G. sulfurreducens* (the same strain we used) transfers electrons to humic acids and similar electron acceptors; hence, it is already established that *Geobacter* spp. can transfer electrons through multi-heme cytochromes in the same manner.

If this proposed pathway is true, how *Geobacter* competes with sludge? Which step is rate limiting?

We previously indicated with two references on line 166 (refs 49 and 50 in the manuscript) and show in the **Fig. 3** schematic that *Geobacter* competes with sludge since it can utilize acetate (acetate was shown to be generated in **Supplementary Fig. 3** for *M. acetivorans* engineered to produce Mcr). The most likely rate-limiting step in our proposed pathway is the initial consumption of methane by the engineered *M. acetivorans* strain.

Is it possible to locate *Geobacter* in solution or biofilm?

Our DNA sequencing of the genomic DNA using the Illumina platform (**Fig. 2**) has already demonstrated clearly the presence of *Geobacter* in the solution of the MFC; i.e., we show that this strain is maintained in the reactors for up to 65 days. As for *Geobacter* in the biofilm, there was no significant biofilm formed by any of the strains (**Supplementary Fig. 5**).

To avoid confusion, we have increased the clarity of **Fig. 3** by revising it to make the acetate and electron transport more clear by adding color.

5. Can you comment on methane mass transfer in liquid and solution acidification in the system?

We also previously demonstrated similar culturing systems whereby a methane headspace is kept over a liquid media consisting of HS or a similar medium to successfully convert methane to acetate⁹ or lactate¹⁰ using a strain of *M. acetivorans*. In these systems, the pH of the liquid medium remained around 7 before and after growth. We thus indicate that there are no major issues regarding methane mass transfer and solution acidification with our system.

6. While it is hard to repeat the calculation in Coulombic efficiency as no raw data was given, but 90±10% of CE is very impressive and to be frankly a little bit too good to be true, especially since mixed sludge was added. Will be good to double check every step of this calculation to make sure it is accurate.

We agree the Coulombic efficiency (CE) from our methane MFCs including *M. acetivorans* producing Mcr from ANME, *G. sulfurreducens*, and methane-acclimated sludge is high, but it is reasonable, since similar CEs of 90%¹¹ and 85%¹² have been reported in MFCs. The high CE found here is due to a number of factors. CEs vary with the cathodic reaction,

and typically increase in MFCs with current density as there is time for aerobic heterotrophs to use more oxygen as an electron acceptor for acetate oxidation than that possible at lower current densities. However, here, the main reason for the high CE is just that there is no competition for the substrate (methane). That is, if it is not used for current generation then there is no other mechanism by which it is oxidized. At the end of the first paragraph of the Discussion (line 150), we now explain this:

“The high CE of this system, compared to other systems where oxygen from the cathode can be used, or other electron acceptors may be present in the solution, reflects the lack of other suitable electron acceptors in the medium other than the anode.”

In order to more carefully explain the basis of our calculations, we have included an example calculation of the CE in the supporting information---, and as suggested we have double checked every step.

References

1. Lovley, D.R. & Blunt-Harris, E.L. Role of humic-bound iron as an electron transfer agent in dissimilatory Fe(III) reduction. *Appl. Environ. Microbiol.* **65**, 4252-4254 (1999).
2. Smith, J.A., Nevin, K.P. & Lovley, D.R. Syntrophic growth via quinone-mediated interspecies electron transfer. *Front. Microbiol.* **6**, 121 (2015).
3. Straub, K.L. & Schink, B. Evaluation of electron-shuttling compounds in microbial ferric iron reduction. *FEMS Microbiol. Lett.* **220**, 229-233 (2003).
4. Scott, D.T., McKnight, D.M., Blunt-Harris, E.L., Kolesar, S.E. & Lovley, D.R. Quinone moieties act as electron acceptors in the reduction of humic substances by humics-reducing microorganisms. *Environ. Sci. Technol.* **32**, 2984-2989 (1998).
5. Lovley, D.R., Coates, J.D., Blunt-Harris, E.L., Phillips, E.J.P. & Woodward, J.C. Humic substances as electron acceptors for microbial respiration. *Nature* **382**, 445-448 (1996).
6. Liu, F., Rotaru, A.E., Shrestha, P.M., Malvankar, N.S., Nevin, K.P. & Lovley, D.R. Promoting direct interspecies electron transfer with activated carbon. *Energy Env. Sci.* **5**, 8982-8989 (2012).
7. Rotaru, A.E., Shrestha, P.M., Liu, F., Shrestha, M., Shrestha, D., Embree, M., Zengler, K., Wardman, C., Nevin, K.P. & Lovley, D.R. A new model for electron flow during anaerobic digestion: direct interspecies electron transfer to *Methanosaeta* for the reduction of carbon dioxide to methane. *Energy Env. Sci.* **7**, 408-415 (2014).
8. Rotaru, A.E., Shrestha, P.M., Liu, F., Markovaite, B., Chen, S., Nevin, K.P. & Lovley, D.R. Direct interspecies electron transfer between *Geobacter metallireducens* and *Methanosarcina barkeri*. *Appl. Environ. Microbiol.* **80**, 4599-4605 (2014).
9. Soo, V.W., McAnulty, M.J., Tripathi, A., Zhu, F., Zhang, L., Hatzakis, E., Smith, P.B., Agrawal, S., Nazem-Bokae, H., Gopalakrishnan, S., Salis, H.M., Ferry, J.G., Maranas, C.D., Patterson, A.D. & Wood, T.K. Reversing methanogenesis to capture methane for liquid biofuel precursors. *Microb. Cell Fact.* **15**, 11 (2016).
10. McAnulty, M.J., Poesarla, V.G., Li, J., Soo, V.W., Zhu, F. & Wood, T.K. Metabolic engineering of *Methanosarcina acetivorans* for lactate production from methane. *Biotechnol. Bioeng.* **114**, 852-861 (2017).
11. Zhang, X., He, W., Ren, L., Stager, J., Evans, P.J. & Logan, B.E. COD removal characteristics in air-cathode microbial fuel cells. *Bioresour. Technol.* **176**, 23-31 (2015).
12. Devasahayam, M. & Masih, S.A. Microbial fuel cells demonstrate high coulombic efficiency applicable for water remediation. *Indian J. Exp. Biol.* **50**, 430-438 (2012).

REVIEWERS' COMMENTS:

Reviewer #2 (Remarks to the Author):

The authors did a good job answering the questions and I can recommend publication.